# Do 2D GANs Know 3D Shape? Unsupervised 3D shape reconstruction from 2D Image GANs

**Xingang Pan**[1]      **Bo Dai**[2]      **Ziwei Liu**[2]      **Chen Change Loy**[2]      **Ping Luo**[3]

[1]The Chinese University of Hong Kong      [2]S-Lab, Nanyang Technological University
px117@ie.cuhk.edu.hk      {bo.dai, ziwei.liu, ccloy}@ntu.edu.sg
[3]The University of Hong Kong
pluo@cs.hku.hk

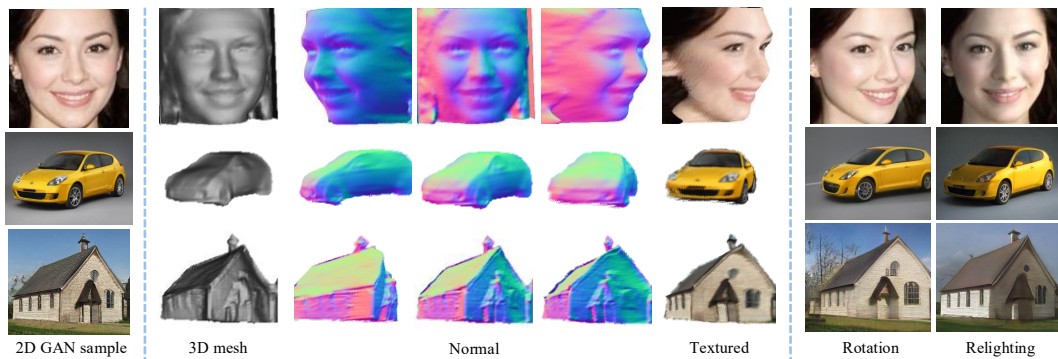

| 2D GAN sample | 3D mesh | Normal | Textured | Rotation | Relighting |

Figure 1: The first column shows images generated by off-the-shelf 2D GANs trained on RGB images only, while the rest show that our method can unsupervisedly reconstruct 3D shape (viewed in 3D mesh, surface normal, and texture) given a single 2D image by exploiting the geometric cues contained in GANs. The last two columns depicts 3D-aware image manipulation effects (rotation and relighting) enabled by our framework. More results are provided in the Appendix.

## Abstract

Natural images are projections of 3D objects on a 2D image plane. While state-of-the-art 2D generative models like GANs show unprecedented quality in modeling the natural image manifold, it is unclear whether they implicitly capture the underlying 3D object structures. And if so, how could we exploit such knowledge to recover the 3D shapes of objects in the images? To answer these questions, in this work, we present the first attempt to directly mine 3D geometric cues from an off-the-shelf 2D GAN that is trained on RGB images only. Through our investigation, we found that such a pre-trained GAN indeed contains rich 3D knowledge and thus can be used to recover 3D shape from a single 2D image in an unsupervised manner. The core of our framework is an iterative strategy that explores and exploits diverse *viewpoint* and *lighting* variations in the GAN image manifold. The framework does not require 2D keypoint or 3D annotations, or strong assumptions on object shapes (*e.g.* shapes are symmetric), yet it successfully recovers 3D shapes with high precision for human faces, cars, buildings, *etc*. The recovered 3D shapes immediately allow high-quality image editing like relighting and object rotation. We quantitatively demonstrate the effectiveness of our approach compared to previous methods in both 3D shape reconstruction and face rotation. Our code is available at *https://github.com/XingangPan/GAN2Shape*.

## 1 Introduction

Generative adversarial networks (GANs) (Goodfellow et al., 2014) are capable of modeling the 2D natural image manifold (Zhu et al., 2016) of diverse object categories with high fidelity. Recall the fact that natural images are actually the projections of 3D objects to the 2D plane, an ideal 2D image

Figure 2: **Framework outline.** Starting with an initial ellipsoid 3D shape (viewed in surface normal), our approach renders various *'pseudo samples'* with different viewpoints and lighting conditions. GAN inversion is applied to these samples to obtain the *'projected samples'*, which are used as the ground truth of the rendering process to refine the initial 3D shape. This process is repeated until more precise results are obtained.

manifold should be able to reflect some underlying 3D geometrical properties. For example, it is shown that a GAN could shift the object in its generated images (*e.g.*, human faces) in a 3D rotation manner, as a direction in the GAN image manifold may correspond to viewpoint change (Shen et al., 2020). This phenomenon motivates us to ask - "*Is it possible to reconstruct the 3D shape of a single 2D image by exploiting the 3D-alike image manipulation effects produced by GANs?*" Despite its potential to serve as a powerful method to learn 3D shape from unconstrained RGB images, this problem remains much less explored.

Some previous attempts (Lunz et al., 2020; Henzler et al., 2019; Szabó et al., 2019) also adopt GANs to learn 3D shapes from images, but they rely on explicitly modeling 3D representation and rendering during training (*e.g.* 3D voxels, 3D models). Due to either heavy memory consumption or additional training difficulty brought by the rendering process, the qualities of their generated samples notably lag behind their 2D GAN counterparts. Another line of works (Wu et al., 2020; Goel et al., 2020; Tulsiani et al., 2020; Li et al., 2020) for unsupervised 3D shape learning generally learns to infer the viewpoint and shape for each image in an 'analysis by synthesis' manner. Despite their impressive results, these methods often assume object shapes are symmetric (symmetry assumption) to prevent trivial solutions, which is hard to generalize to asymmetric objects such as 'building'.

We believe that existing pre-trained 2D GANs, without above specific designs, already capture sufficient knowledge for us to recover the 3D shapes of objects from 2D images. Since the 3D structure of an instance could be inferred from images of the same instance with multiple *viewpoint* and *lighting* variations, our insight is that we may create these variations by exploiting the image manifold captured by 2D GANs. However, the main challenge is to discover well-disentangled semantic directions in the image manifold that control viewpoint and lighting in an unsupervised manner, as manually inspect and annotate the samples in the image manifold is laborious and time-consuming.

To tackle the above challenge, we observe that for many objects such as faces and cars, a convex shape prior like ellipsoid could provide a hint on the change of their viewpoints and lighting conditions. Inspired by this, given an image generated by GAN, we employ an ellipsoid as its initial 3D object shape, and render a number of unnatural images, called *'pseudo samples'*, with various randomly-sampled viewpoints and lighting conditions as shown in Fig. 2. By reconstructing them using the GAN, these pseudo samples could guide the original image towards the sampled viewpoints and lighting conditions in the GAN manifold, producing a number of natural-looking images, called *'projected samples'*. These projected samples could be adopted as the ground truth of the differentiable rendering process to refine the prior 3D shape (*i.e.* an ellipsoid). To achieve more precise results, we further regard the refined shape as the initial shape and repeat the above steps to progressively refine the 3D shape.

With the proposed approach, namely *GAN2Shape*, we show that existing 2D GANs trained on images only are sufficient to accurately reconstruct the 3D shape of a single image for many object categories such as human faces, cars, buildings, *etc*. Our method thus is an effective approach for unsupervised 3D shape reconstruction from unconstrained 2D images *without* any 2D keypoint or 3D annotations. With an improved GAN inversion strategy, our method works not only for GAN samples, but also for real natural images. On the BFM benchmark (Paysan et al., 2009), our method outperforms a recent strong baseline designed specifically for 3D shape learning (Wu et al., 2020). We also show high-quality 3D-aware image manipulations using the semantic latent directions discovered by our approach, which achieves more accurate human face rotation than other competitors.

Our contributions are summarized as follows. **1)** We present the first attempt to reconstruct the 3D object shapes using GANs that are pre-trained on 2D images only. Our work shows that 2D GANs

inherently capture rich 3D knowledge for different object categories, and provides a new perspective for 3D shape generation. **2)** Our work also provides an alternative unsupervised 3D shape learning method, and does not rely on the symmetry assumption of object shapes. **3)** We achieve highly photo-realistic 3D-aware image manipulations including rotation and relighting without using any external 3D models.

## 2 RELATED WORK

**Generative Adversarial Networks.** Generative Adversarial Networks (GANs) (Goodfellow et al., 2014; Xiangli et al., 2020) have achieved great success in 2D natural image modeling. After a series of improvements, the state-of-the-art GANs like StyleGAN (Karras et al., 2019; 2020b) and BigGAN (Brock et al., 2019) could synthesis images with high fidelity and resolution. These GANs are composed of 2D operations, like 2D convolutions. Recently, there is a surge of interest to model 3D-aware image distributions with 3D GAN architectures and 3D representations (Nguyen-Phuoc et al., 2019; Wu et al., 2016; Szabó et al., 2019; Henzler et al., 2019; Lunz et al., 2020). However, due to either heavy memory consumption or increased difficulty in training, there are still gaps between their image qualities and those of 2D GANs.

Therefore, in this work, we are interested in investigating whether we can recover the 3D object shape by directly mining the knowledge of 2D GANs. Our resulting method could be viewed as a memory-efficient way for 3D shape and texture generation, which shares the benefits of both 2D and 3D generative models, *e.g.*, high quality and resolution samples, and a 3D interpretation of contents.

**3D-aware Generative Manipulation.** Given a pre-trained GAN, there has been works trying to discover the latent space directions that manipulate the image content in a 3D-controllable manner. InterFaceGAN (Shen et al., 2020) learns to rotate human faces by learning latent direction from manual annotations. In contrast, SeFa (Shen & Zhou, 2020) and GANSpace (Härkönen et al., 2020) discover meaningful latent directions without supervision, where some of them is coupled with content pose. Recently, StyleRig (Tewari et al., 2020), DiscoFaceGAN (Deng et al., 2020), and RotateRender (Zhou et al., 2020) achieves 3D-aware manipulation on human face by using an external 3D human face model, *i.e.*, 3DMM (Blanz & Vetter, 1999), as a strong 3D shape prior.

Unlike these methods, our method could accurately manipulate on both pose and lighting factors *without borrowing external 3D models or supervisions*. This allows our method to generalize beyond human faces, like cats and cars.

**Unsupervised 3D Shape Learning.** Unsupervised learning of 3D shapes from raw, monocular view images has attracted increasing attention recently, as it relieves the heavy burden on manual annotations. The key challenge of this problem is the lack of multiple view or lighting images for a single instance. To tackle this problem, a number of works leverage external 3D shape models or weak supervisions like 2D key-points. For example, Gecer et al. (2019); Kanazawa et al. (2018a); Sanyal et al. (2019); Shang et al. (2020) reconstruct the 3D shape of human faces or bodies using 3DMM (Blanz & Vetter, 1999) or SMPL (Loper et al., 2015) as a prior knowledge. And Tran & Liu (2018); Kanazawa et al. (2018b) adopt the guidance of 2D key-points to learn the 3D shape of face or objects. The works of Tulsiani et al. (2020); Goel et al. (2020) further discard the need of former supervisions, but still rely on initial category-specific shape templates.

Some representative works that rely only on natural 2D images include Sahasrabudhe et al. (2019); Wu et al. (2020); Li et al. (2020)[1], which learn to disentangle the viewpoint and shape for each image via autoencoders that explicitly model the rendering process. However, these methods still rely on certain regularization to prevent trivial solutions. For example, Sahasrabudhe et al. (2019) adopts an autoencoder with a small bottleneck embedding to constrain the deformation field, which tend to produce blur and imprecise results. Wu et al. (2020); Li et al. (2020) adopt the assumption that many object shapes are symmetric, thus their inferred shapes sometimes miss the unsymmetric aspects of the objects in images.

Our method in this work provides an alternative solution for the fore-mentioned 'single view' challenge, which is to obtain multiple view and lighting images by exploiting the image manifold modeled by 2D GANs. Our method does not necessarily rely on the symmetry assumption on object

---

[1] We include methods that require silhouettes in this part.

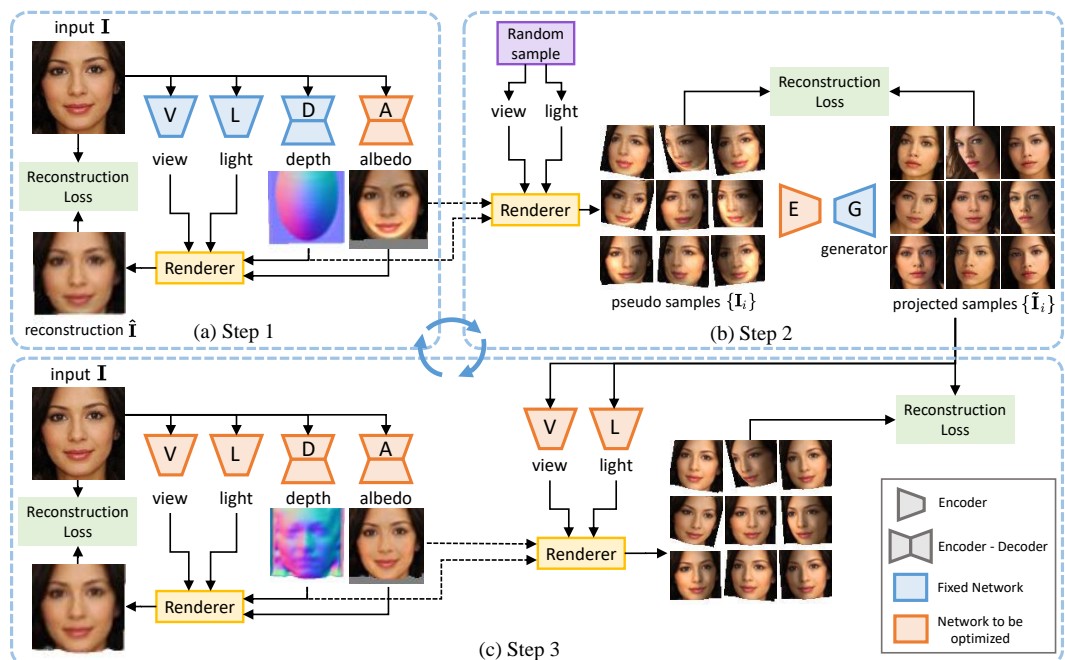

Figure 3: **Method overview.** (a) Given a single image, Step 1 initializes the depth with ellipsoid (viewed in surface normal), and optimizes the albedo network $A$. (b) Step 2 uses the depth and albedo to render 'pseudo samples' with various random viewpoint and lighting conditions, and conducts GAN inversion to them to obtain the 'projected samples'. (c) Step 3 refines the depth map by optimizing $(V, L, D, A)$ networks to reconstruct the projected samples. The refined depth and models are used as the new initialization to repeat the above steps.

shape, thus better captures the asymmetric aspects of objects. In experiments, we demonstrate superior performance over a recent state-of-the-art method Unsup3d (Wu et al., 2020).

## 3 METHODOLOGY

**Preliminaries on Generative Adversarial Networks.** We first provide some preliminaries on GANs before discussing how we exploit GANs for 3D shape recovery. GANs (Goodfellow et al., 2014) have achieved great successes in modeling natural images. A GAN consists of a generator $G$ that maps a latent vector $z$ to an image $x \in \mathbb{R}^{3 \times H \times W}$, and a discriminator $D$ that distinguishes between generated images and real ones. $z$ is drawn from a prior distribution such as the multivariate normal distribution $\mathcal{N}(0, I)^d$. GANs trained on natural images are able to continuously sample realistic images, thus could be used as an approximation of the natural image manifold (Zhu et al., 2016).

In this work, our study is mainly based on a representative GAN, StyleGAN2 (Karras et al., 2020b). The generator of StyleGAN2 consists of two parts, a mapping network $F : \mathcal{Z} \mapsto \mathcal{W}$ that maps the latent vector $z$ in the input latent space $\mathcal{Z}$ to an intermediate latent vector $w \in \mathcal{W}$, and a synthesis network that maps $w$ to the output image. In the following sections, we refer to the synthesis network as $G$. The image manifold of StyleGAN2 thus could be defined as $\mathbb{M} = \{G(w)|w \in \mathcal{W}\}$. Such a two-stage structure makes it easier to perform GAN-inverson on top of StyleGAN2. *i.e.*, searching for a $w$ that best reconstructs a real target image (Abdal et al., 2019).

### 3.1 FROM 2D GANS TO 3D SHAPES

In order to mine the viewpoint and lighting cues in the GAN image manifold, we need to explicitly model these factors. Here we follow the photo-geometric autoencoding design in Wu et al. (2020). For an image $\mathbf{I} \in \mathbb{R}^{3 \times H \times W}$, we adopt a function that is designed to predict four factors $(d, a, v, l)$,

including a *depth map* $\boldsymbol{d} \in \mathbb{R}^{H \times W}$, an *albedo image* $\boldsymbol{a} \in \mathbb{R}^{3 \times H \times W}$, a *viewpoint* $\boldsymbol{v} \in \mathbb{R}^6$, and a *light direction* $\boldsymbol{l} \in \mathbb{S}^2$. This function thus is implemented with four corresponding sub-networks $(D, A, V, L)$, as shown in Fig. 3 (a). The four factors are recomposed to form the original image $\mathbf{I}$ via a rendering process $\Phi$, which consists of *lighting* $\Lambda$ and *reprojection* $\Pi$ steps as follows:

$$\begin{aligned} \hat{\mathbf{I}} &= \Phi(\boldsymbol{d}, \boldsymbol{a}, \boldsymbol{v}, \boldsymbol{l}) \\ &= \Pi(\Lambda(\boldsymbol{d}, \boldsymbol{a}, \boldsymbol{l}), \boldsymbol{d}, \boldsymbol{v}) \end{aligned} \tag{1}$$

Here $\Lambda$ performs shading with albedo $\boldsymbol{a}$, depth map $\boldsymbol{d}$, and light direction $\boldsymbol{l}$, while $\Pi$ performs viewpoint change and generates the image viewed from viewpoint $\boldsymbol{v}$. The viewpoint change is realized with a differentiable renderer (Kato et al., 2018). We put more details in the Appendix.

**Step 1: Using a Weak Shape Prior.** As our goal is to explore images of the same instance in the GAN image manifold that have novel viewpoints and lighting conditions, we would like to seek for a hint on the change of viewpoint and lighting to guide the exploration. To achieve this, we make the observation that many objects including faces and cars have a somewhat convex shape prior. We could thus resort to an ellipsoid as a weak prior to create 'pseudo samples' with varying viewpoints and lightings as the hint.

Specifically, as shown in Fig.3 (a), given an image sample $\mathbf{I}$ from the GAN, we initialize the depth map $\boldsymbol{d}$ to have an ellipsoid shape. With an off-the-shelf scene parsing model (Zhao et al., 2017), we could position the ellipsoid to be roughly aligned with the object in the image. Then the albedo network $A$ is optimized with the reconstruction objective $\mathcal{L}(\mathbf{I}, \hat{\mathbf{I}})$, where $\hat{\mathbf{I}}$ is calculated via Eq. (1), and $\mathcal{L}$ is a weighted combination of L1 loss and perceptual loss (Johnson et al., 2016). The viewpoint $\boldsymbol{v}$ and lighting $\boldsymbol{l}$ are initialized with a canonical setting, *i.e.*, $\boldsymbol{v}_0 = 0$ and lighting is from the front. This allows us to have an initial guess about the four factors $\boldsymbol{d}_0, \boldsymbol{a}_0, \boldsymbol{v}_0, \boldsymbol{l}_0$, with $\boldsymbol{d}_0$ being the weak shape prior.

**Step 2: Sampling and Projecting to the GAN Image Manifold.** With the above shape prior as an initialization, we are able to create 'pseudo samples' by sampling a number of random viewpoints $\{\boldsymbol{v}_i | i = 1, 2, ..., m\}$ and lighting directions $\{\boldsymbol{l}_i | i = 1, 2, ..., m\}$. Specifically, we use $\{\boldsymbol{v}_i\}$ and $\{\boldsymbol{l}_i\}$ along with $\boldsymbol{d}_0$ and $\boldsymbol{a}_0$ to render pseudo samples $\{\mathbf{I}_i\}$ via Eq. (1). $\{\boldsymbol{v}_i\}$ and $\{\boldsymbol{l}_i\}$ are randomly sampled from some prior distributions, like the multi-variate normal distribution. As shown in Fig.3 (b), although these pseudo samples have unnatural distortions and shadows, they provide cues on how the face rotates and how the light changes.

In order to leverage such cues to guide novel viewpoint and lighting direction exploration in the GAN image manifold, we perform GAN inversion to these pseudo samples, *i.e.*, reconstruct them with the GAN generator $G$. Specifically, we train an encoder $E$ that learns to predict the intermediate latent vector $\boldsymbol{w}_i$ for each sample. Unlike prior works that directly predict the latent vector, in our setting the latent vector $\boldsymbol{w}$ for the original sample $\mathbf{I}$ is known, thus the encoder $E$ only need to predict an offset $\Delta \boldsymbol{w}_i = E(\mathbf{I}_i)$ that encodes the difference between the pseudo sample $\mathbf{I}_i$ and the original sample $\mathbf{I}$, which is much easier. Thus the optimization goal is:

$$\boldsymbol{\theta}_E = \arg\min_{\boldsymbol{\theta}_E} \frac{1}{m} \sum_{i=0}^{m} \mathcal{L}'\left(\mathbf{I}_i, G\big(E(\mathbf{I}_i) + \boldsymbol{w}\big)\right) + \lambda_1 \|E(\mathbf{I}_i)\|_2 \tag{2}$$

where $m$ is the number of samples, $\boldsymbol{\theta}_E$ is the parameter of the encoder $E$, $\lambda_1$ is the regularization coefficient, and $\mathcal{L}'$ is a distance metric for images. Following Pan et al. (2020), we adopt the L1 distance of the discriminator features as the distance metric $\mathcal{L}'$, which is shown to work better for GAN samples. The regularization term $\|E(\mathbf{I}_i)\|_2$ is to prevent the latent offset to grow too large during training.

As shown in Fig.3 (b), to achieve better reconstruction, the projected samples $\{\tilde{\mathbf{I}}_i = G(E(\mathbf{I}_i) + \boldsymbol{w})\}$ should have similar viewpoint and lighting changes as the pseudo samples $\{\mathbf{I}_i\}$. In the meanwhile, the generator $G$ could regularize the projected samples to lie in the natural image manifold, thus fixing the unnatural distortions and shadows in the pseudo samples. In the next step, we discuss how to exploit these projected samples to infer the 3D shape.

**Step 3: Learning the 3D Shape.** The above projected samples $\{\tilde{\mathbf{I}}_i\}$ provide images of multiple viewpoint and lighting conditions while having nearly the same object content, *e.g.*, face identity. Such *'single instance multiple view&lighting'* setting makes it possible to learn the underlying

3D shape with aforementioned photo-geometric autoencoding model. Specifically, as illustrated in Fig.3 (c), the viewpoint network $V$ and the lighting network $L$ predict instance-specific view $\tilde{v}_i$ and lighting $\tilde{l}_i$ for each sample $\tilde{\mathbf{I}}_i$, while the depth network $D$ and albedo network $A$ output the shared depth $\tilde{v}$ and albedo $\tilde{a}$ with the original sample $\mathbf{I}$ as input. The four networks are jointly optimized with the following reconstruction objective:

$$\boldsymbol{\theta}_D, \boldsymbol{\theta}_A, \boldsymbol{\theta}_V, \boldsymbol{\theta}_L = \underset{\boldsymbol{\theta}_D, \boldsymbol{\theta}_A, \boldsymbol{\theta}_V, \boldsymbol{\theta}_L}{\arg\min} \frac{1}{m} \sum_{i=0}^{m} \mathcal{L}\Big(\tilde{\mathbf{I}}_i, \Phi\big(D(\mathbf{I}), A(\mathbf{I}), V(\tilde{\mathbf{I}}_i), L(\tilde{\mathbf{I}}_i)\big)\Big) + \lambda_2 \mathcal{L}_{smooth}\big(D(\mathbf{I})\big)$$

(3)

where $\boldsymbol{\theta}_D, \boldsymbol{\theta}_A, \boldsymbol{\theta}_V, \boldsymbol{\theta}_L$ are the parameters of network $D, A, V, L$ respectively, and $\mathcal{L}_{smooth}\big(D(\mathbf{I})\big)$ is a smoothness term defined the same way as in (Zhou et al., 2017). Note that we also reconstruct the original sample $\mathbf{I}$, *i.e.*, use $\mathbf{I}$ as an additional $\tilde{\mathbf{I}}_i$, to prevent the infered factors to deviate from the original sample. Here we do not use the sampled $\{v_i\}$ and $\{l_i\}$ in step 2, as the viewpoints and lighting directions of the projected samples may not accurately match the pseudo samples, thus we learn to predict them with $V$ and $L$ instead.

The above reconstruction loss would decrease only when the underlying depth, albedo, and all samples' viewpoints and lighting directions are correctly inferred. As shown in Fig.3 (c), the object shape, parameterized by depth, does shift toward the true underlying shape, *e.g.*, shape of a human face. And the unnatural distortions and shadows that exist in the pseudo samples are mitigated in the reconstructions.

**Iterative Self-refinement.** While the above three steps produce a more accurate 3D object shape than the prior shape, it may miss some details like the wrinkles on the face. This is because the projected samples $\{\tilde{\mathbf{I}}_i\}$ may not accurately preserve all semantics of the original instance in $\mathbf{I}$, as the pseudo samples are obtained from an approximated weak shape prior. Fortunately, we can repeat the cycle of these three steps multiple times by regarding the refined 3D shape and trained networks in the previous cycle as the initialized shape and networks in a new cycle. This allows us to refine the 3D shape and projected samples progressively. In our experiments, we use four cycles of these three steps unless otherwise stated.

## 3.2 DISCUSSION

**Regularizing the Latent Offset.** In the Eq. 2 of Step 2, the latent offset $\Delta \boldsymbol{w}_i = E(\mathbf{I}_i)$ is only regularized with a weak L2 norm term, thus the resulting latent vector $\Delta \boldsymbol{w}_i + \boldsymbol{w}$ may fall out of the intermediate latent distribution $\mathcal{W}$ of the GAN. This could possibly cause the projected samples $\{\tilde{\mathbf{I}}_i\}$ to inherit some unnatural distortions in the pseudo samples $\{\mathbf{I}_i\}$, thus stronger regularization on $\Delta \boldsymbol{w}_i$ is required. To address this issue, we propose a new regularization strategy, which is to restrict $\Delta \boldsymbol{w}_i$ to be a valid offset created by the mapping network $F$ as $\Delta \boldsymbol{w}_i = F(E(\mathbf{I}_i)) - F(\mathbf{0})$. We provide more discussion on this design in the Appendix.

**Joint Training of Multiple Instances.** Note that our discussion above involves only a single target image $\mathbf{I}$, *i.e.*, our method is used in an instance-specific manner. However, our pipeline could also be jointly trained on multiple instances to achieve a better generalization ability and a faster convergence speed on new samples. The extension to joint training is straightforward, whose details are included in the Appendix. The key difference between training with multiple instances and a single instance is that, since there are multiple target instances with diverse viewpoint and lighting variations, the absolute latent offset $E(\mathbf{I}_i)$ in Eq. 2 should be replaced with a relative latent offset $E(\mathbf{I}_i) - E(\mathbf{I})$, which could generalize better across different target instances.

## 4 EXPERIMENTS

**Implementation Details.** We first evaluate our method on 3D shape recovery, and then show its application to 3D-aware image manipulation. The datasets used include CelebA (Liu et al., 2015), BFM (Paysan et al., 2009), a combined cat dataset (Zhang et al., 2008; Parkhi et al., 2012), LSUN Car (Yu et al., 2015), and LSUN Church (Yu et al., 2015), all of which are unconstrained RGB images. For BFM, we use the same one as in Wu et al. (2020). We adopt StyleGAN2 pre-trained on these datasets in our experiments. We recommend readers to refer to the Appendix for more implementation details and qualitative results.

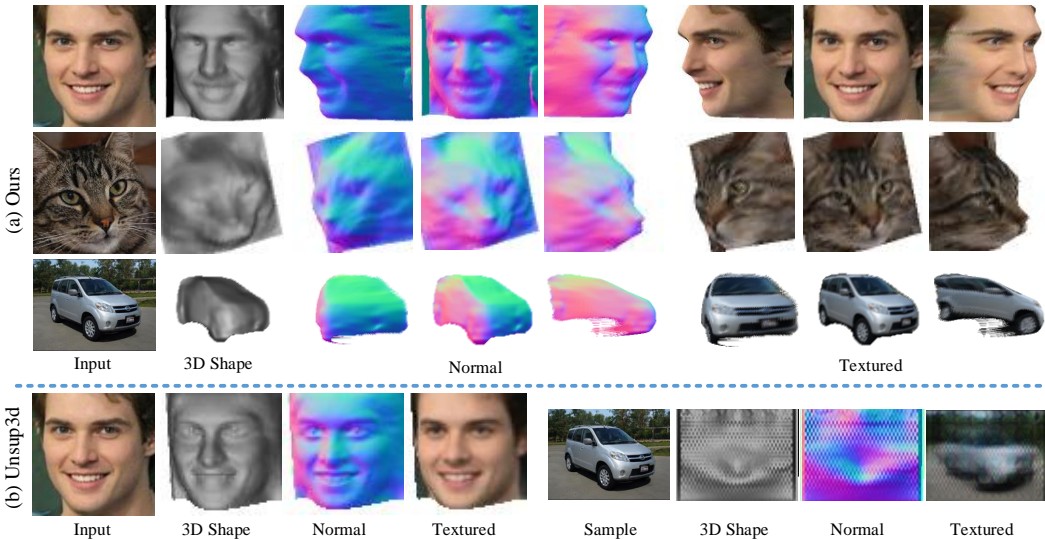

Figure 4: **Qualitative comparisons.** (a) shows the reconstructed 3D mesh, surface normal, and textured mesh of our method. (b) shows the results of Unsup3d (Wu et al., 2020). We see that results in (a) are more accurate and realistic.

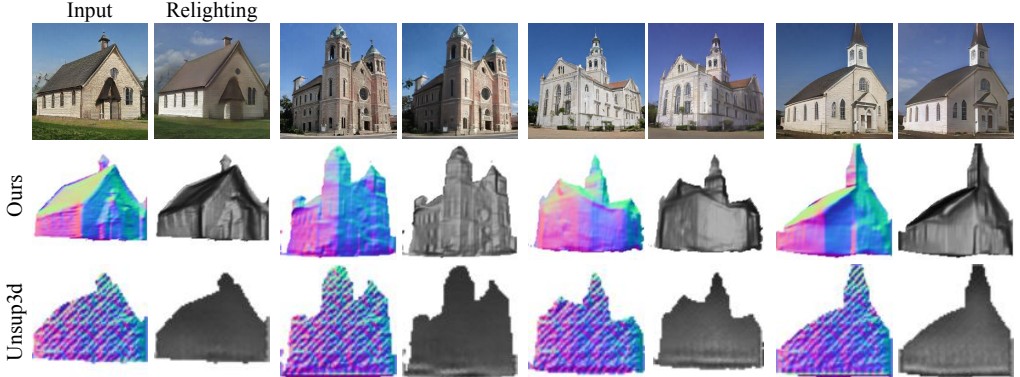

Figure 5: **Qualitative comparison on buildings.** The first row shows the input image and our relighting effects described in Sec.4.2. The second row shows the recovered shape (viewed in surface normal and mesh) of our method, while the last row shows the results of Unsup3d.

## 4.1 UNSUPERVISED 3D SHAPE RECONSTRUCTION

**Qualitative Evaluation.** The qualitative results of our method and Unsup3d (Wu et al., 2020) are shown in Fig.1, Fig.4, and Fig.5. Our method recovers the 3D shapes with high-quality across human faces, cats, cars, and buildings. For example, the wrinkles on human faces and the edges and flat surfaces on cars are well captured. The results of Unsup3d, while good, tend to miss some asymmetric aspects of human faces, *e.g.*, its predicted eye directions always point to the front. Besides, as shown in Fig.4 (b) and Fig.5, Unsup3d has difficulty in generalizing to datasets with asymmetric objects or large viewpoint variations like buildings and cars, while our method does not.

**Quantitative Evaluation.** Quantitative comparison is conducted on the BFM dataset. Following (Wu et al., 2020), we report the scale-invariant depth error (SIDE) and mean angle deviation (MAD) scores as the evaluation metrics. In order to evaluate on the test set images instead of GAN generated samples, we need to first conduct GAN inversion, *i.e.*, finding the intermediate latent vectors $\{w\}$ that best reconstruct these images. We adopt the method in Karras et al. (2020b), which achieves satisfactory reconstruction results, as shown in the first row of Fig.6. To achieve better generalization, our method is pre-trained on only 200 training images before testing (the training set has 160k images in total). Due to the additional testing time brought by the GAN inversion and instance-specific training steps, we report results on the first 500 images of the test set.

Table 1: **Comparisons on the BFM dataset.** We report SIDE and MAD errors. 'Symmetry' indicates whether the symmetry assumption on object shape is used. We outperform others on both metrics.

| No. | Method | Symmetry | SIDE ($\times 10^{-2}$)↓ | MAD (deg.)↓ |
|-----|--------|----------|--------------------------|-------------|
| (1) | Supervised | N | 0.419 | 10.83 |
| (2) | Const. null depth | / | 2.723 | 43.22 |
| (3) | Average g.t. depth | / | 1.978 | 22.99 |
| (4) | Unsup3d (Wu et al., 2020) | Y | 0.807 | 16.34 |
| (5) | Ours (w/o regularize) | Y | 0.925 | 16.42 |
| (6) | Ours | Y | **0.756** | **14.81** |
| (7) | Unsup3d (Wu et al., 2020) | N | 1.334 | 33.79 |
| (8) | Ours | N | **1.023** | **17.09** |

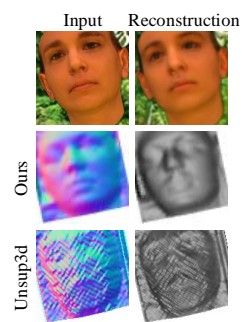

Figure 6: Results without symmetry assumption.

Table 2: **Effects of different shape prior.** We report results of the original ellipsoid shape, asymmetric shape with ellipsoid for the left half and sphere for the right half, shape with its position shifted by 1/6 and 1/4 image width, weaker shape prior whose height is half of the original one, and no shape prior. Qualitative results can be found in Fig.14 in the Appendix.

| Shape prior | Origin | Asymmetric | Shift w/6 | Shift w/4 | Weak | Flat |
|-------------|--------|------------|-----------|-----------|------|------|
| SIDE ($\times 10^{-2}$)↓ | 0.756 | 0.769 | 0.767 | 0.775 | 0.764 | 1.021 |
| MAD (deg.)↓ | 14.81 | 14.95 | 14.93 | 15.07 | 14.97 | 20.46 |

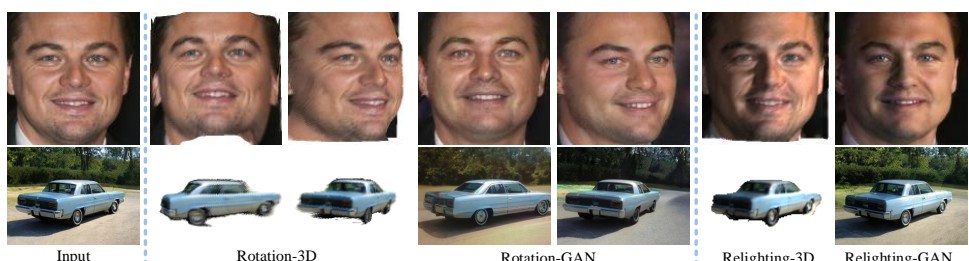

Input | Rotation-3D | Rotation-GAN | Relighting-3D | Relighting-GAN

Figure 7: **3D-aware image manipulation**, including rotation and relighting. We show results obtained via both 3D mesh and GANs. The input of the first row is a real natural image. Our method achieves photo-realistic manipulation effects obeying the objects' underlying 3D structures.

Although the above GAN inversion step could introduce additional error due to imperfect reconstruction, our method still noticeably outperforms Unsup3d and other trivial baselines, as shown in Tab.1. It is worth noting that our method does not necessarily rely on the symmetric assumption. To demonstrate this, we also report results of our method and Unsup3d when both are learned without the symmetric assumption. As No.(7)(8) in Tab.1 and Fig.6 show, in this setting our method significantly outperforms Unsup3d, which tends to produce messy results. Furthermore, No.(5) provides the results without using the latent offset regularization described in Sec.3.2, where the performance decreases notably, demonstrating the effectiveness of the proposed regularization technique.

We further study the effects of using different shape prior. The setting is the same as Tab.1, and the results are shown in Tab.2. Some variations like asymmetric, shifted, and weaker shape prior only slightly affect the performance, showing that the result is not very sensitive to the shape prior. But the results would get worse for flat shape, as it could not indicate viewpoint and lighting change.

## 4.2 3D-AWARE IMAGE MANIPULATION

**Object Rotation and Relighting.** After training, the recovered 3D shape and the encoder $E$ immediately allow various 3D-aware image manipulation effects. In Fig.7, we show two manipulation effects, including object rotation and relighting. For each effect, we show results rendered using the recovered 3D shape and albedo, and their corresponding projections in the GAN manifold via the encoder $E$. The results rendered from 3D shapes strictly follow the underlying structure of the object, while their projections in GAN are more photo-realistic. As mentioned before, our method is applicable to real natural images via GAN inversion, and the first row in Fig.7 is an example, where our method also works well.

Table 3: **Identity-preserving face rotation.** We compare with HoloGAN, GANSpace, and SeFa. The metrics are identity distances measured as angles in the ArcFace feature embeddings.

| Method | error_mean (deg.)↓ | error_max (deg.)↓ |
|---|---|---|
| HoloGAN | 47.38 | 69.24 |
| GANSpace | 41.17 | 58.93 |
| SeFa | 41.79 | 60.73 |
| Ours (3D) | **28.93** | **43.02** |
| Ours (GAN) | 39.85 | 57.21 |

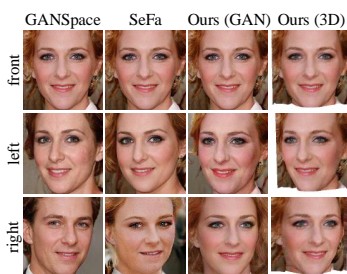

Figure 8: **Qualitative comparison on face rotation.** "Ours (GAN)" and "Ours (3D)" indicate results generated by GAN and rendered from 3D mesh respectively. The face identities in the baseline methods tend to drift during rotation.

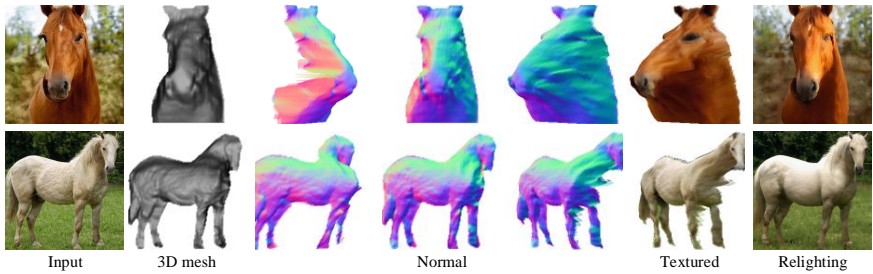

Figure 9: **Qualitative results on the LSUN Horse dataset** (Yu et al., 2015).

**Identity-preserving Face Rotation.** We compare our method with other unsupervised methods that also achieve face rotation with GANs, including HoloGAN (Nguyen-Phuoc et al., 2019), GANSpace (Härkönen et al., 2020), and SeFa (Shen & Zhou, 2020). Specifically, for each method, we randomly sample 100 face images, and for each face, we shift its yaw angle from -20 degree to 20 degree, obtaining 20 samples with different poses. A face identity detection model ArcFace (Deng et al., 2019) is then used to evaluate how much the face identities change during rotation. More details are included in the Appendix. Since our method explicitly recover the 3D face shapes, results rendered from these recovered 3D shapes significantly outperform other counterparts, as shown in Tab.3 and Fig.8. In the alternative attempt where we directly operate in the GAN manifold following the projected directions of the recovered 3D shapes, face identities are also better preserved.

**Discussion.** We have shown that our method is applicable to many object categories. However, for objects with more sophisticated shapes like horses, a simple convex shape prior may not well reflect the viewpoint and lighting variations, and thus the 3D shapes could not be very accurately inferred. The second row of Fig.9 provides one example. Another limitation of our method is that our 3D mesh is parameterized by a depth map, which could not model the back-side shape of objects, as the first row of Fig.9 shows. This could possibly be addressed by adopting a better parameterization of the 3D mesh. Despite these limitations, our method still captures some aspects of the horses' shapes, like the rough shape of the head and belly, and achieves reasonable relighting effects.

## 5 CONCLUSION

We have presented the first method that directly leverages off-the-shelf 2D GANs to recover 3D object shapes from images. We found that existing 2D GANs inherently capture sufficient knowledge to recover 3D shapes for many object categories, including human faces, cats, cars, and buildings. Based on a weak convex shape prior, our method could explore the viewpoint and lighting variations in the GAN image manifold, and exploit these variations to refine the underlying object shape in an iterative manner. We have further shown the application of our method on 3D-aware image manipulations including object rotation and relighting. Our results uncover the potential of 2D GANs in modeling the underlying 3D geometry of 2D image manifold.

**Acknowledgment.** This research was conducted in collaboration with SenseTime. This work is supported by NTU NAP and A*STAR through the Industry Alignment Fund - Industry Collaboration Projects Grant. This work is also supported by the HK General Research Fund No.27208720.

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

# A APPENDIX

In this appendix, we provide more qualitative results and the implementation details.

## A.1 QUALITATIVE EXAMPLES

Fig.10 provides more qualitative results of our method, including 3D shape reconstruction and 3D-aware image manipulations. The effects of iterative training is shown in Fig.11, where the object shapes become more precise at latter training stages. Fig.12 shows that our method works well without using the symmetry assumption. Fig.13 shows that our method achieves reasonable results for many challenging cases.

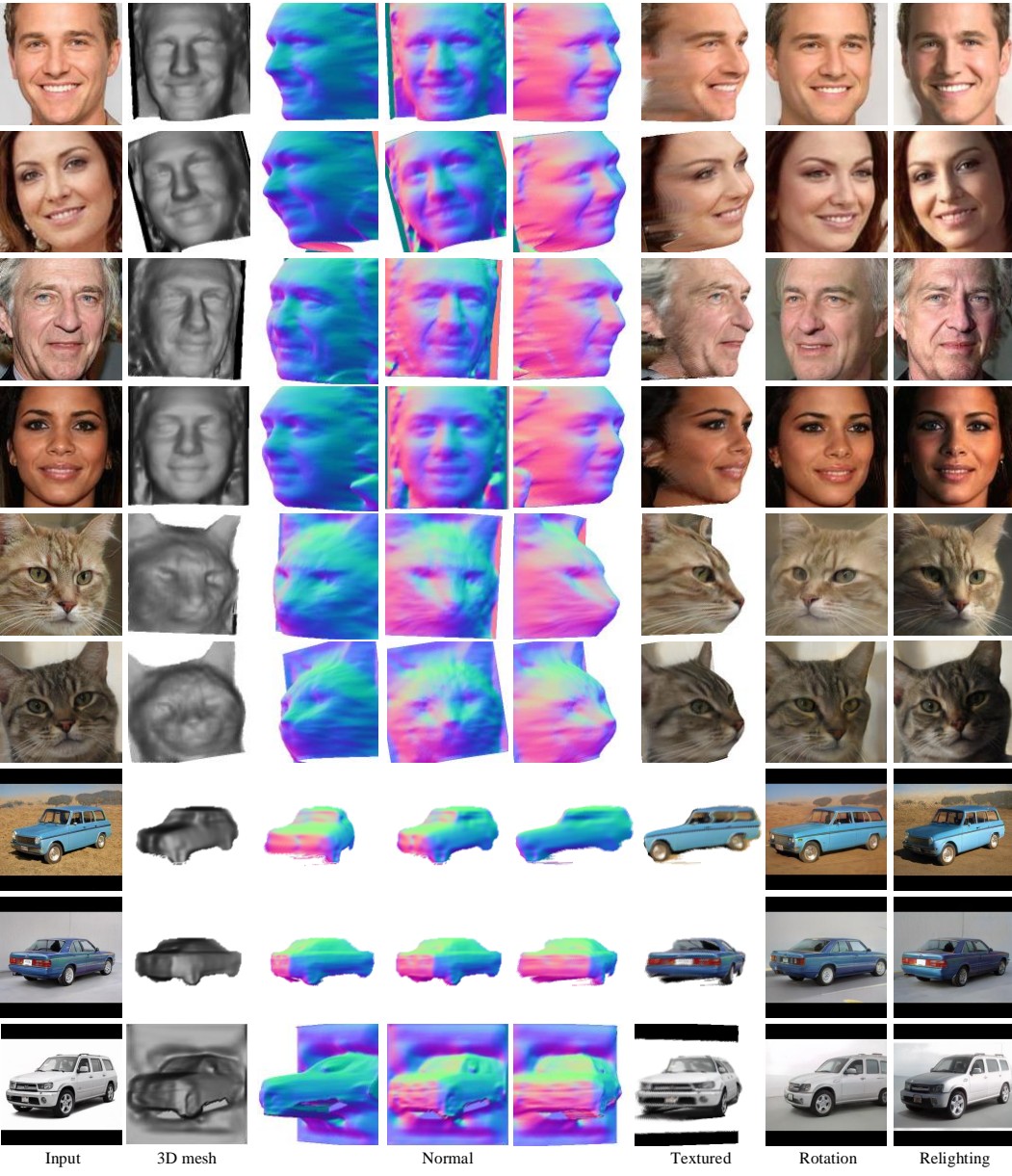

| Input | 3D mesh | | Normal | | Textured | Rotation | Relighting |

Figure 10: This is an extension of Fig.1. The last row shows a car example without removing the background.

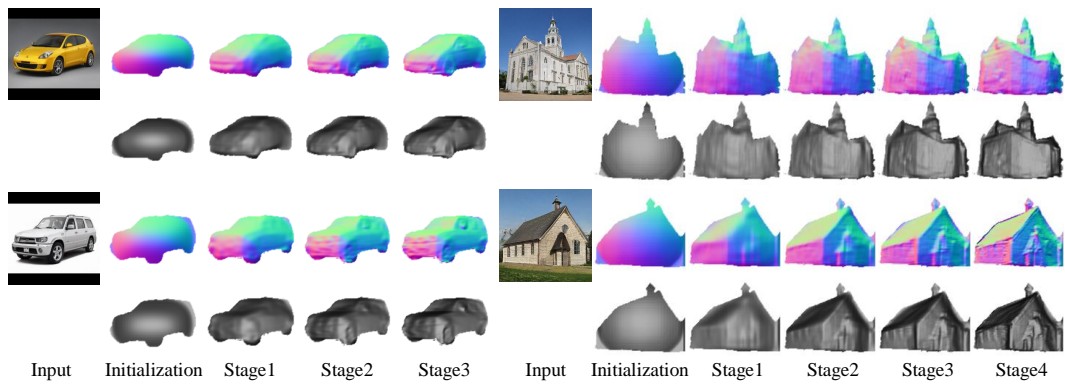

Input   Initialization  Stage1   Stage2   Stage3    Input  Initialization  Stage1  Stage2  Stage3  Stage4

Figure 11: **Effects of iterative training.** The reconstructed object shapes get more precise at latter training stages.

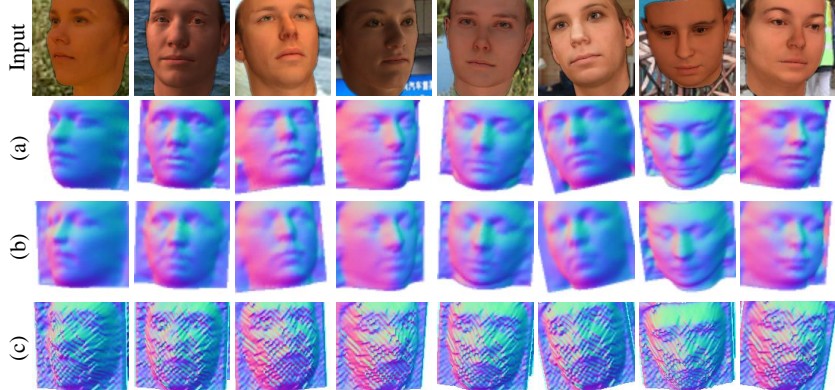

Figure 12: (a) Our results with the symmetry assumption. (b) Our results without the symmetry assumption. (c) The results of Unsup3d Wu et al. (2020) without the symmetry assumption. Our method achieves competitive results even without using the symmetry assumption.

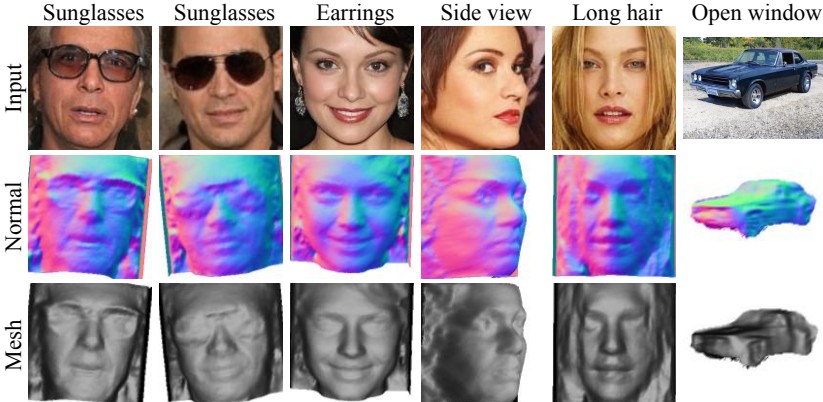

Figure 13: **Results on challenging cases**. Our method achieves reasonable results for most cases.

**Effects of shape prior.** Fig.14 shows the qualitative results of using different shape priors. Unlike the results in Tab.2, here joint training is not adopted to better showcase the effects of different shape. When the ellipsoid is shifted by 1/6 image width or is asymmetric, the shape could be progressively refined during the iterative training process. But the results would get worse if the shift is too large or a flat shape is used, because the viewpoint and lighting changes could not be revealed with these shapes.

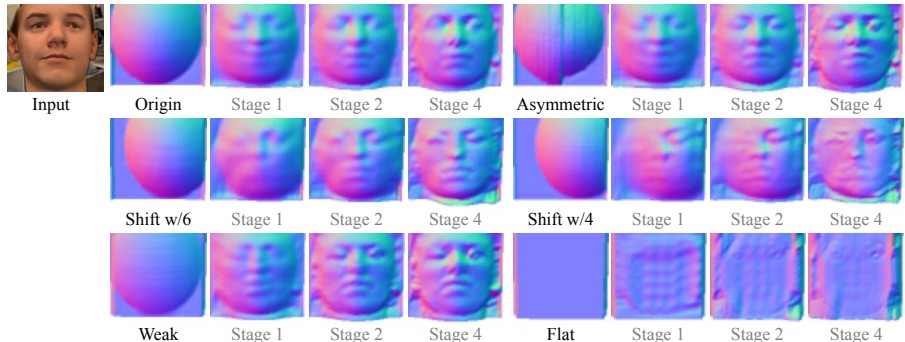

Figure 14: **Effects of different shape prior**. We show results of the original ellipsoid shape, asymmetric shape with ellipsoid for the left half and sphere for the right half, shape with its position shifted by 1/6 and 1/4 image width, weaker shape prior whose height is half of the original one, and flat shape.

Table 4: **Effects of dataset size.** We report quantitative results on the BFM dataset with different number of training samples.

| Dataset size | 160k | 40k | 10k | 3k | 1k |
|---|---|---|---|---|---|
| SIDE ($\times 10^{-2}$)↓ | 0.756 | 0.765 | 0.784 | 0.794 | 0.945 |
| MAD (deg.)↓ | 14.81 | 14.90 | 15.10 | 15.28 | 16.84 |

**Effects of dataset size.** Fig.4 shows the effects of dataset size used for training the GAN. It can be seen that the performance only drops marginally even when the dataset size is reduced from 160k to 3k. And the performance get notably worse when the dataset size reduces to 1k, as the image quality of the GAN starts to deteriorate given insufficient data. This could potentially be addressed by recent data augmentation strategy for GANs (Karras et al., 2020a).

## A.2 IMPLEMENTATION DETIALS

### A.2.1 PHOTO-GEOMETRIC AUTOENCODING

The photo-geometric autoencoding model we used mainly follows the work of Wu et al. (2020). Here we describe the $\Lambda$ and $\Pi$ in Eq.1.

$\mathbf{J} = \Lambda(\boldsymbol{d}, \boldsymbol{a}, \boldsymbol{l})$ produces the illuminated texture with the albedo $\boldsymbol{a}$, depth map $\boldsymbol{d}$, and light direction $\boldsymbol{l}$. Specifically, we first calculate the normal map $\boldsymbol{n}$ of the depth map $\boldsymbol{d}$, which describes the normal direction of the surface for each pixel position $(i, j)$. Then the illuminated texture is calculate with $J_{ij} = (k_s + k_d \max 0, \langle l, n_{ij} \rangle) \cdot a_{ij}$, where $k_s$ and $k_d$ are the scalar weight for the ambient and diffuse terms, and are predicted by the lighting network $V$. The lighting direction is described as $\boldsymbol{l} = (l_x, l_y, 1)^T / (l_x^2 + l_y^2 + 1)^{0.5}$, where $l_x$ and $l_y$ are also the predictions of the lighting network.

Given the above illuminated texture $\mathbf{J}$, the reprojection function $\Pi$ would map it to a new viewpoint $\boldsymbol{v}$ as $\hat{\mathbf{I}} = \Pi(\mathbf{J}, \boldsymbol{d}, \boldsymbol{v})$. To do this, we denote the 3D point in the reference frame of the camera as $\boldsymbol{P} = (P_x, P_y, P_z) \in \mathbb{R}^3$, which is projected to pixel $\boldsymbol{p} = (i, j, 1)$ via:

$$\boldsymbol{p} \propto \boldsymbol{KP}, \quad \boldsymbol{K} = \begin{bmatrix} f & 0 & c_i \\ 0 & f & c_j \\ 0 & 0 & 1 \end{bmatrix}, \quad \begin{cases} c_i = \frac{W-1}{2}, \\ c_j = \frac{H-1}{2}, \\ f = \frac{W-1}{2 \tan \frac{\theta_{FOV}}{2}} \end{cases} \tag{4}$$

where $\theta_{FOV}$ is the field of view of the perspective camera, and is assume to be $\theta_{FOV} \approx 10°$. Given a depth map $\boldsymbol{d}$ which describes a depth value $d_{ij}$ for each pixel $(i, j)$, we would have $\boldsymbol{P} = d_{ij} \cdot \boldsymbol{K}^{-1} \boldsymbol{p}$.

The viewpoint $\boldsymbol{v} \in \mathbb{R}^6$ represents the rotation angles ($\boldsymbol{v}_{1:3}$) and translations ($\boldsymbol{v}_{4:6}$) along $x$, $y$ and $z$ axes. $\boldsymbol{v}$ thus describes an Euclidean transform $(\boldsymbol{R}, \boldsymbol{T}) \in SE(3)$, which transforms 3D points to the predicted viewpoint. The mapping from a pixel $(i, j)$ in the original view to the pixel $(i', j')$ in the

predicted view could then be described by the warping function $\eta_{d,v} : (i, j) \mapsto (i', j')$ as:

$$\boldsymbol{p}' \propto \boldsymbol{K}(d_{ij} \cdot \boldsymbol{R}\boldsymbol{K}^{-1}\boldsymbol{p} + \boldsymbol{T}) \tag{5}$$

where $\boldsymbol{p}' = (i', j', 1)$. Finally, $\Pi$ applies this warp to the illuminated texture $\mathbf{J}$ to obtain the reprojected image $\hat{\mathbf{I}} = \Pi(\mathbf{J}, \boldsymbol{d}, \boldsymbol{v})$ as $\hat{I}_{i'j'} = J_{ij}$, where $(i, j) = \eta_{d,v}^{-1}(i', j')$. Here $\eta_{d,v}^{-1}$ is the backward warping of $\eta_{d,v}$, which is achieved via a differentiable renderer Kato et al. (2018) in the same way as Wu et al. (2020).

### A.2.2 Regularizing the Latent Offset

As described in Section 3.2, in order to keep the projected samples $\{\tilde{\mathbf{I}}_i\}$ in the natural image manifold, stronger regularization on the predicted latent offset $\Delta\boldsymbol{w}_i$ is needed. One possible solution is to let the encoder predict the offset in the original latent space, *i.e.*, the one before the mapping network $F$, then the output of the generator would be $G\big(F(E(\mathbf{I}_i) + \boldsymbol{z})\big)$, where $\boldsymbol{z}$ is the original latent vector corresponding to $\mathbf{I}$. However, one issue of this solution is that, compared with the intermediate latent space $\mathcal{W}$, it is much more difficult to perform GAN inversion to the original latent space $\mathcal{Z}$ to obtain the latent vector $\boldsymbol{z}$ of a real natural image. This would impede this method to be applied to real natural images other than GAN samples.

To address this issue, in this work we propose a new regularization strategy for the latent offset $\Delta\boldsymbol{w}_i$, which is to restrict it to be a valid offset created by the mapping network $F$ as follows:

$$\begin{aligned} \Delta\boldsymbol{w}_i &= F(\Delta\boldsymbol{z}_i) - F(\mathbf{0}) \\ &= F(E(\mathbf{I}_i)) - F(\mathbf{0}) \end{aligned} \tag{6}$$

which regards $\Delta\boldsymbol{w}_i$ as the shift in the $\mathcal{W}$ space by adding perturbation around zero in the $\mathcal{Z}$ space. Note that the point zero is a natural choice as it is the mean point in the prior distribution $\mathcal{N}(0, I)^d$. In order to make the regularization strength adjustable, one can further view the mapping network $F$ as two consecutive parts $F = F_1 \circ F_2$, and regularize $\Delta\boldsymbol{w}_i$ with only $F_1$ as below:

$$\Delta\boldsymbol{w}_i = F_1(E(\mathbf{I}_i) + F_2(\mathbf{0})) - F(\mathbf{0}) \tag{7}$$

The depth $d$ of $F_1$ thus controls the strength of regularization. As shown in Tables 10 11 12, for the BFM dataset, we use the full mapping network (*i.e.*, $d = 8$). For other datasets with more complex images, the regularization of a full mapping network is too strong and sometimes lead to sub-optimal results, we thus set $d = 2$.

### A.2.3 Training Detials

In this section, we provide the network architectures, hyper-parameters, and other details in our experiments.

**Model Architectures**. For a fair comparison on the BFM dataset, the images used in the photogeometric model are resized to $64^2$ resolution, and the $D, A, L, V$ network have the same architecture as Wu et al. (2020). For other datasets, the image resolution is resized to $128^2$, and the network architectures are added by additional layers accordingly, as shown in Tab.5 and Tab.6 . For the GAN encoder net $E$, we adopt a ResNet-alike architecture (He et al., 2016), as presented in Tab.7 and Tab.8 . The abbreviations for the network layers are described below:

Conv($c_{in}, c_{out}, k, s, p$): convolution with $c_{in}$ input channels, $c_{out}$ output channels, kernel size $k$, stride $s$, and padding $p$.

Deconv($c_{in}, c_{out}, k, s, p$): deconvolution with $c_{in}$ input channels, $c_{out}$ output channels, kernel size $k$, stride $s$, and padding $p$.

GN($n$): group normalization (Wu & He, 2018) with $n$ groups.

LReLU($\alpha$): leaky ReLU (Maas et al., 2013) with a negative slope of $\alpha$.

Upsample($s$): nearest-neighbor upsampling with a scale of $s$.

Avg_pool($s$): average pooling with a stride of $s$.

ResBlock($c_{in}, c_{out}$): residual block as defined in Tab.8.

Table 6: Network architecture for depth net $D$ and albedo net $A$. $c_{out}$ is 1 for $D$ and 3 for $A$.

| Encoder | Output size |
|---|---|
| Conv(3, 32, 4, 2, 1) + GN(8) + LReLU(0.2) | 64 |
| Conv(32, 64, 4, 2, 1) + GN(16) + LReLU(0.2) | 32 |
| Conv(64, 128, 4, 2, 1) + GN(32) + LReLU(0.2) | 16 |
| Conv(128, 256, 4, 2, 1) + GN(64) + LReLU(0.2) | 8 |
| Conv(256, 512, 4, 2, 1) + LReLU(0.2) | 4 |
| Conv(512, 256, 4, 1, 0) + ReLU | 1 |

| Decoder | Output size |
|---|---|
| Deconv(256,512,4,1,0) + ReLU | 4 |
| Conv(512,512,3,1,1) + ReLU | 4 |
| Deconv(512,256,4,2,1) + GN(64) + ReLU | 8 |
| Conv(256,256,3,1,1) + GN(64) + ReLU | 8 |
| Deconv(256,128,4,2,1) + GN(32) + ReLU | 16 |
| Conv(128,128,3,1,1) + GN(32) + ReLU | 16 |
| Deconv(128,64,4,2,1) + GN(16) + ReLU | 32 |
| Conv(64,64,3,1,1) + GN(16) + ReLU | 32 |
| Deconv(64,32,4,2,1) + GN(8) + ReLU | 64 |
| Conv(32,32,3,1,1) + GN(8) + ReLU | 64 |
| Upsample(2) | 128 |
| Conv(32,32,3,1,1) + GN(8) + ReLU | 128 |
| Conv(32,32,5,1,2) + GN(8) + ReLU | 128 |
| Conv(32,$c_{out}$,5,1,2) + Tanh | 128 |

Table 5: Network architecture for view-point net $V$ and lighting net $L$. The output channel size $c_{out}$ is 6 for $V$ and 4 for $L$.

| Encoder | Output size |
|---|---|
| Conv(3, 32, 4, 2, 1) + ReLU | 64 |
| Conv(32, 64, 4, 2, 1) + ReLU | 32 |
| Conv(64, 128, 4, 2, 1) + ReLU | 16 |
| Conv(128, 256, 4, 2, 1) + ReLU | 8 |
| Conv(256, 512, 4, 2, 1) + ReLU | 4 |
| Conv(512, 512, 4, 1, 0) + ReLU | 1 |
| Conv(512, $c_{out}$, 1, 1, 0) + Tanh | 1 |

Table 7: Network architecture of GAN encoder net $E$ for $128^2$ resolution input images. For $64^2$ resolution input, the last ResBlock is removed and the following channels are cut down by a half.

| Encoder | Output size |
|---|---|
| Conv(3, 32, 4, 2, 1) + ReLU | 64 |
| ResBlock(32, 64) | 32 |
| ResBlock(64, 128) | 16 |
| ResBlock(128, 256) | 8 |
| ResBlock(256, 512) | 4 |
| Conv(512, 1024, 4, 1, 0) + ReLU | 1 |
| Conv(1024, 512, 1, 1, 0) | 1 |

Table 8: Network architecture for the ResBlock($c_{in}$, $c_{out}$) in Tab.7. The output of Residual path and Identity path are added as the final output.

| Residual path |
|---|
| ReLU + Conv($c_{in}$, $c_{out}$, 3, 2, 1) |
| ReLU + Conv($c_{out}$, $c_{out}$, 3, 1, 1) |

| Identity path |
|---|
| Avg_pool(2) |
| Conv($c_{in}$, $c_{out}$, 1, 1, 0) |

Table 9: Hyper-parameters.

| Parameter | Value/Range |
|---|---|
| Optimizer | Adam |
| Learning rate | $1 \times 10^{-4}$ |
| Depth map | (0.9, 1.1) |
| Ellipsoid | (0.91, 1.02) |
| $\lambda_1$ (in Eq.2) | 0.01 |
| $\lambda_2$ (in Eq.3) | 0.01 |

Table 10: Hyper-parameters for LSUN Car and LSUN Church datasets.

| Parameter | Value | Symmetry |
|---|---|---|
| Number of pseudo samples $m$ | 3200 | |
| Depth $d$ of $F_1$ | 2 | |
| Batchsize | 32 | |
| Number of stages | 4 | |
| Step 1 iterations (1st stage) | 700 | |
| Step 2 iterations (1st stage) | 700 | |
| Step 3 iterations (1st stage) | 600 | N |
| Step 1 iterations (other stages) | 200 | |
| Step 2 iterations (other stages) | 500 | |
| Step 3 iterations (other stages) | 400 | N |

**Hyper-parameters.** The hyper-parameters of our experiments are provided in Tables 9 10 11 12. The batch size here is the batch size used in the Step 2 and Step 3 of our method. And the number of stages indicates how many times the Step 1-3 are repeated.

**Joint pre-training.** Note that for the CelebA, Cat, and BFM dataset, we adopt joint pre-training on 200 samples to achieve better generalization when fine-tuning on new samples. Specifically, for each iteration of each step, we randomly choose a batch of samples from the 200 samples, and the training is the same as that of a single sample except that the gradient calculated from each sample are averaged after back-propagation.

**Symmetry assumption.** In our method, the symmetry assumption on object shapes is an optional choice, and could further improve the results for human face and cat datasets. Thus we also adopt this assumption for CelebA, Cat, and BFM dataset unless otherwise stated. Note that the results

Table 11: Hyper-parameters for CelebA and Cat datasets.

| Joint Pre-train | Value | Symmetry |
|---|---|---|
| Number of samples | 200 | |
| Number of pseudo samples $m$ | 256 | |
| Depth $d$ of $F_1$ | 2 | |
| Batchsize | 64 | |
| Number of stages | 4 | |
| Step 1 iterations (1st stage) | 1500 | |
| Step 2 iterations (1st stage) | 1500 | |
| Step 3 iterations (1st stage) | 1500 | Y |
| Step 1 iterations (other stages) | 700 | |
| Step 2 iterations (other stages) | 1000 | |
| Step 3 iterations (other stages) | 1000 | Y |
| Fine-tune | Value | Symmetry |
| Number of pseudo samples $m$ | 1600 | |
| Depth $d$ of $F_1$ | 2 | |
| Batchsize | 16 | |
| Number of stages | 4 | |
| Step 1 iterations (1st stage) | 600 | |
| Step 2 iterations (1st stage) | 600 | |
| Step 3 iterations (1st stage) | 400 | Y |
| Step 1 iterations (other stages) | 200 | |
| Step 2 iterations (other stages) | 500 | |
| Step 3 iterations (other stages) | 300 | N |

Table 12: Hyper-parameters for BFM dataset.

| Joint Pre-train | Value | Symmetry |
|---|---|---|
| Number of samples | 200 | |
| Number of pseudo samples $m$ | 480 | |
| Depth $d$ of $F_1$ | 8 | |
| Batchsize | 96 | |
| Number of stages | 8 | |
| Step 1 iterations (1st stage) | 1500 | |
| Step 2 iterations (1st stage) | 1500 | |
| Step 3 iterations (1st stage) | 1500 | Y |
| Step 1 iterations (other stages) | 700 | |
| Step 2 iterations (other stages) | 1000 | |
| Step 3 iterations (other stages) | 1000 | Y |
| Fine-tune | Value | Symmetry |
| Number of pseudo samples $m$ | 1600 | |
| Depth $d$ of $F_1$ | 8 | |
| Batchsize | 16 | |
| Number of stages | 1 | |
| Step 1 iterations | 500 | |
| Step 2 iterations | 500 | |
| Step 3 iterations | 200 | Y |

without the symmetry assumption are also provided in Figures 6 12 , Tab.12 and other datasets, which is also satisfactory. Unlike Unsup3d (Wu et al., 2020), our method does not need to predict an additional confidence map for the symmetry, as we can drop the symmetry assumption after the first training stage, as shown in Tab.11 .

**Sampling viewpoints and lighting directions.** In the Step 2 of our method, we need to randomly sample a number of viewpoints and lighting directions from some prior distributions. For viewpoint, we calculate the mean and covariance statistics of viewpoints estimated from the BFM, CelebA, and Cat dataset using the viewpoint prediction network of (Wu et al., 2020). These statistics are used to sample the random viewpoints from multi-variate normal distributions for the corresponding datasets. For LSUN Car and Church datasets, we simply use the statistics estimated from the FFHQ dataset (Karras et al., 2019), and we found it works well.

For lighting directions, we sample from the following uniform distributions: $l_x \sim \mathcal{U}(x_{min}, x_{max})$, $l_y \sim \mathcal{U}(y_{min}, y_{max})$, $k_d \sim \mathcal{U}(d_{min}, d_{max})$, and $k_s = \alpha k_d$, where $[x_{min}, x_{max}, y_{min}, y_{max}, d_{min}, d_{max}, \alpha] = [-0.9, 0.9, -0.3, 0.8, -0.1, 0.7, -0.4]$ for BFM dataset and $[-1, 1, -0.2, 0.8, -0.1, 0.6, -0.6]$ for other datasets. Note the sampled $k_d$ and $k_s$ are added to the predictions of the lighting network $L$ as the final diffuse and ambient term. This setting tends to increase the diffuse term and decrease the ambient term, as the diffuse term could better reflect the structure of the objects.

**Masking out the background.** In this work, we care about the main object rather than the background. Thus, for GANs trained on the LSUN Car and LSUN Church datasets that contain large background area, we could use an off-the-shelf scene parsing model PSPNet50 (Zhao et al., 2017) to parse the object areas as a pre-processing for better visualization. For an input image, the scene parsing model is used only once to get the object mask for the original view, thus it does not provide signals on the 3D shape. For novel views, the mask is warped with the current shape accordingly, and is used to mask out the image background in Eq.2 and Eq.3 . For other datasets we do not mask out the background.

**Face rotation evaluation.** In the identity-preserving face rotation experiments, we use a face identity detection model ArcFace (Deng et al., 2019) to evaluate the shift of face identity during rotation. For each human face sample, we first rotate its yaw angle from -20 degree to 20 degrees to obtain 20 samples with different poses. The face angle is obtained with the viewpoint prediction network from Wu et al. (2020). We then calculate the identity distances between the front-view face and other faces, and report both their mean and max values in Tab.11 . For face identity prediction model, the identity distances are represented as angles between the final feature embeddings. The above mean and max values are averaged across 100 samples as the final results.

