# OpenReview forum: "Do 2D GANs Know 3D Shape? Unsupervised 3D Shape Reconstruction from 2D Image GANs"
_ICLR.cc/2021/Conference — ICLR 2021 Oral_

### Official Review · AnonReviewer2 · 2020-10-29
**Good work that estimates 3D shapes w/o supervision with the help of pre-trained GANs**

**Rating:** 8
**Confidence:** 4

**Review:**

This paper proposes an iterative method that jointly estimates viewpoints, light directions, depth, and albedo from single images, by projecting intermediate renderings to the nautral image manifold. Intuitively, the method works by generating, with pre-trained GANs, multiple views of the same object under different lightings, and then inferring 3D shapes from those variants. The key idea is to use pre-trained 2D GANs to make such data generation photorealistic. The authors also demonstrate 3D edits, such as 3D rotation and relighting, that one can perform after running their model.

I like this paper because
(1) it presents the novel idea of "generating", by GAN inversion, photorealistic multi-view, multi-light data of the given real object, from which the 3D shape can then be estimated;
(2) extensive evaluations were performed to demonstrate the high quality achieved; and
(3) one can perform 3D edits, such as 3D rotation and relighting, on top of the model outputs. Having explicit 3D understanding for relighting makes a lot of sense to me, and this paper presents a new angle of doing so by GAN inversion.

In terms of drawbacks, this paper would benefit from the following experiments or clarifications:
(1) How crucial is the size of the dataset for GAN pretraining? For example, if the dataset is small, not covering many of the face poses, I imagine the GAN projections may not always look realistic, thereby causing the shape estimation to degrade. Such failure cases or studies should be shown so that the reader understands what impact the dataset bias or size has on the final results. An example would be a plot of shape reconstruction error w.r.t. the dataset size or face pose coverage.
(2) How robust is the algorithm to the shape initialization (ellipsoid shape)? More importantly, what about its location -- what if the "off-the-shelf scene parsing model" fails so that the ellipsoid is placed off the main object? If the model fails because of bad initializations, what do the failure modes look like? Are the shapes completely garbage, or something that looks like the initialization?
(3) Why is the viewing direction parameterized in R6 (I presume the start and end XYZs)? Shouldn't it be in S2, just like the light direction? If they are in R6, then results on zooming in/out should be included. Otherwise, there seems to be no point in defining them in R6.

---

> ### Author Response · Authors · 2020-11-21
> **Reply to Reviewer #2**
>
> Thank you for your constructive comments. Below we address the concerns. We also include our revision in the paper (content in blue), and an animated demo in the supplementary material.
>
> *****Q1: How crucial is the size of the dataset for GAN pretraining?*****
> A1: We have studied the effects of using different dataset sizes, the results are listed as below (also included in Tab.4 in the revised Appendix):
>
> | Dataset size  | 160k   |  40k    |  10k   |   3k   |   1k |
> | -------------------   | ------------------ | ------------------ | ------------------ | ------------------ |  ------------------ |
> | SIDE              | 0.773 |  0.782 |  0.805 | 0.816 | 0.962 |
> | MAD              | 15.32  | 15.41 |  15.62 | 15.78 | 17.36 |
>
> It can be seen that the performance only drops marginally even when the dataset size is reduced from 160k to 3k. The performance gets notably worse when the dataset size reduces to 1k, mainly due to the image quality of the GAN starting to deteriorate given insufficient data. This could potentially be addressed by recent data augmentation strategy for GANs [1].
>
> [1] Karras, Tero, et al. "Training generative adversarial networks with limited data." NeurIPS2020.
>
> *****Q2: How robust is the algorithm to the shape initialization?*****
> A2: We have included the results of using a number of different shape initializations in Tab.2 and Fig.14 in the revised version, which show that our result is not very sensitive to the shape initialization. As shown in Fig.14, when the ellipsoid is shifted by 1/6 image width or is asymmetric, the shape could be progressively refined during the iterative training process. But the results would get worse if the shift is too large or a flat shape is used, because the viewpoint and lighting changes could not be revealed with these shapes. For example, when the shape initialization is flat, the final result only adds some small random perturbations.
>
> *****Q3: Why is the viewing direction parameterized in R6?*****
> A3: The viewpoint v should be parameterized in R6 because this concept here is not simply a direction, but describes the Euclidean transformation of the camera in the world coordinate [2]. It is also known as the view/camera transformation. Therefore, it needs 6 dimensions, where 3 of them describe rotation angles including pitch, yaw, and roll, and the other 3 of them describe translations along x, y and z axes. S2 describes a direction, but not an Euclidean transformation, and it is also insufficient to describe rotation angle as there is another degree of freedom rotating along the direction.
>
> [2] http://www.codinglabs.net/article_world_view_projection_matrix.aspx

---

### Official Review · AnonReviewer4 · 2020-10-29
**This paper firstly proposes an unsupervised method to recover 3D shape from 2D image using GANs. The results are state-of-the-art and the experiments are comprehensive.**

**Rating:** 7
**Confidence:** 3

**Review:**

Pros:
1. This is the first work that attempts to reconstruct 3D shape from 2D image in an unsupervised way using GANs. The idea is neat: Use networks to predict four 3D parameters and use GAN to generate / synthesize the images corresponding to a set of parameters. Then these synthesized images can be used as pseudo ground truth to train the 3D parameter network.
2. The experiments are comprehensive to support the effectiveness of the proposed pipeline. 2 tasks are evaluated: 3D shape reconstruction and object-aware image manipulation. On 3D shape reconstruction, performances are reported on two datasets and demonstrated it outperforms SoTA method by a large margin. On image manipulation, the visualization results look reasonable and visually better than previous method.

Cons / Questions:
1. The authors claim in the introduction section that this proposed method has advantage over previous method as it doesn't assume symmetry of the instance. But in this proposed method, a symmetrical ellipsoid is used as the shape prior. Is this a stronger implicit assumption than the symmetry assumption? Is there any experiment to explore how the shape prior affects the models' training results? E.g. what if a non-convex shape prior is provided, what if asymmetrical prior is provided?
2. In the introduction section, the authors use `'building' as an example to show that previous method's symmetry assumption cannot work. But in the experiments and comparison with previous works, all data used are symmetrical: human face, animal face, cars, etc. Visualizations on buildings are shown in appendix but there is no quantitative analysis or comparing with current SoTA method.
3. How is the generator initialized? As the generator is always fixed during training, I assume the generator is using some pre-trained network? If so, can we still call this network fully unsupervised?

---

> ### Author Response · Authors · 2020-11-21
> **Reply to Reviewer #4**
>
> Thank you for your constructive comments. Below we address the concerns. We also include our revision in the paper (content in blue), and an animated demo in the supplementary material.
>
> *****Q1: Is ellipsoid a stronger assumption? How the shape prior affects the results?*****
> A1: We believe that ellipsoid is a weaker implicit assumption than the symmetry assumption, because the symmetry assumption would restrict the shape to be symmetric throughout the whole training process, while the ellipsoid is only for initialization and there is NO restriction on the shape during training. Therefore, in our method the predicted object shape could be asymmetric while the methods relying on the symmetry assumption could not.
>
> We have included the results of using a number of different shape priors in Tab.2 and Fig.14 in the revised version, which show that our result is not very sensitive to the shape prior. As shown in Fig.14, even when the shape prior is asymmetric and non-convex, the shape could be progressively refined during the iterative training process and the final result is still accurate. The SIDE and MAD scores of using an asymmetric shape prior are 0.789 and 15.48 respectively, which are comparable to the original performances. While our result is not very sensitive to the shape prior, it should be nearly convex to indicate viewpoint/lighting changes, otherwise it would result in trivial solutions as shown in the “Flat” case in Fig.14.
>
> *****Q2: Comparison with current method on ‘building’ is needed.*****
> A2: We have provided a comparison with Unsup3d on ‘building’ in Fig.5 of the revised paper. Since there lacks a building dataset with ground truth shape annotations, we provide a qualitative comparison, where Unsup3d fails to predict meaningful 3D shapes and our method significantly outperforms it.
>
> *****Q3: How is the generator initialized? Can we call it unsupervised?*****
> A3: As mentioned in Sec.4 “Implementation details”, the GANs are pre-trained on the datasets with the standard adversarial loss. This unconditional GAN training process itself is still unsupervised as no manual annotation is needed and only raw images are involved. Thus we can still call this network unsupervised.

---

### Official Review · AnonReviewer1 · 2020-10-29
**[Official Review]**

**Rating:** 8
**Confidence:** 5

**Review:**

#### Summary ####
This paper studies an interesting inverse-graphics problem. It proposed a novel method to learn 3D shape reconstruction using pre-trained 2D image generative adversarial networks. Given an image containing one single object of interest, it first predicts the graphics code (e.g., viewpoint, lighting, depth, and albedo) by minimizing the reconstruction error using a differentiable renderer. The next step is to render many pseudo samples by randomization in the viewpoint and lighting space, while keeping the predicted depth and albedo fixed. A pre-trained 2D image GAN is further used to project the pseudo samples to the learned data manifold through GAN-Inversion. Finally, these projected samples are added to the set for the next round optimization. Experimental evaluations have been conducted on several categories including face, car, building, and horse.


#### Comments ####
Overall, this is a very interesting paper with good presentations, promising experimental results, and solid quantitative comparisons with the previous work. Reviewer would like to point out the potential weakness of the paper as follows.

W1: Though impressed by the results (especially the proposed method works for horse and building), reviewer suspects the paper only works in a very simplified setting: (1) the GAN was previously trained on a large amount of 2D images of a single category with many variations in identity, viewpoint, and lighting; (2) the initialization (or step 1 in Section 3.1) step seems very critical to the overall performance; and (3) viewpoint and lightning randomization seems have to be hand-tuned. Reviewer would like to see the discussions on the underlying assumptions more explicitly. In addition, reviewer would like to know how does the method generalize to “dirty” data: people with sunglasses, people with noticeable earrings, people partially occluded by wavy long hair, and people with a side view (looks like the input has to be a frontal face image). Same question applies to those non-convex shapes: a convertible car or a car with the window open. Reviewer suspects the method in the current form cannot handle them well.

W2: Some important experimental settings are neither presented nor clarified. For example, it is not clear what is the difference between “Ours (3D)” and “Ours (GAN)”, which should be clarified. For image editing (see Figure 6 and Figure 8), reviewer sees a noticeable change in background color sometimes (e.g., second row in Figure 6). It would be good to give a very detailed explanation of the image editing process (e.g., what’s the input and output format in each stage). As ellipsoid was used to initialize the face shape, reviewer would like to know what was the initialization for other categories such as building and car (see Figure 10).

W3: It would be good to report the time spent on the computation and optimization and how it is compared to the baselines in Table 1 and Table 2. This is very important metric to report as a fair comparison to the previous work.

== Post-rebuttal Comments ==

I am raising my score from 7 to 8, as author responses addressed my comments well (especially answer to W1 and the Figure 13) than expected.

---

> ### Author Response · Authors · 2020-11-21
> **Reply to Reviewer #1**
>
> Thank you for your insightful comments. Below we address the concerns. We also include our revision in the paper (content in blue), and an animated demo in the supplementary material.
>
> *****Q1: Discussion on underlying assumptions and results on “dirty” data.*****
>
> A1: Thanks for the suggestion, we agree that the results on “dirty” data is important to show the capabilities and limitations of the method. To begin with, we would like to make some clarifications.
> (1) The dataset of a single category with many viewpoint and lighting variations may seem simplified but is reasonable, as geometric clues are reflected in these viewpoint and lighting variations. The datasets used in our paper are standard datasets for GAN training and 3D shape learning.
> (2) While the initialized shape plays an important role in the framework, the final result is **not very sensitive** to it, as long as it could reflect viewpoint/lighting changes. We have added the results of using different shape priors in Tab.2.
> (3) Viewpoint and lightning randomization are **not very carefully hand-tuned**. As described in the Sec.A.2.3 of Appendix, we use the same randomization for all experiments, where the viewpoint randomness is estimated from the Celeba dataset.
> To be more specific, the underlying assumption is that a GAN could learn a regularized image manifold such that its output images should obey the geometric properties of the object, and in the meanwhile the learned image manifold could cover the viewpoint and lighting variations of the dataset. Therefore, as long as the initial shape could reflect viewpoint/lighting changes, the reconstruction loss could shift the GAN output similarly while still obeying the object geometry, and that is how the geometric clues are mined from the GAN. Although the viewpoint/lighting randomization used in practice may not precisely match the variations of the target dataset (e.g., if the viewpoint change of a pseudo sample goes beyond the distribution of the dataset, the projected sample could not reach the same viewpoint.), the geometric clues could be mined as long as there is an overlap between the viewpoint/lighting randomness of the dataset and the pseudo samples.
>
> As for the “dirty” data, we have provided the results in Fig.13 of the revision. For faces with sunglasses, earrings, and long hair, the results are not perfect but still reasonable, e.g., the edges of the glasses and hair are captured. Our method also works well for the side view case, but could not capture the open window on a car, as there may not be enough geometric clues in such a subtle area.
>
> *****Q2: Some experimental settings are not clarified.*****
>
> A2: The difference between “Ours (3D)” and “Ours (GAN)” is explained in Sec.4.2 line 3. “Ours (3D)” is rendered using the recovered 3D shape and albedo, while “Ours (GAN)” is obtained by performing GAN-inversion to “Ours (3D)” using the trained encoder E. We have made this clearer in the revision.
>
> Due to limited paper length, the change in background color is explained in Sec A.2.3 “Masking out the background” of the Appendix. For car and building images with a large background, we mask out the background for better visualization. More details on how this process is conducted is also provided in the Appendix. Our code would be released to make the results reproducible.
>
> Ellipsoid is used as the initialization for all object categories including car and building. The shapes in Fig.10 are also initialized with ellipsoid, and the only difference is that the background is masked out for better visualization as mentioned above.
>
> *****Q3: Computation and optimization time.*****
>
> A3: For the results in Tab.1, Unsup3d takes 27 hours to train as it learns a global image-to-shape mapping by jointly training on the whole dataset, while our method conducts instance-specific training to exploit GAN knowledge, which takes 14 minutes for an image. For our method, joint training is an optional choice and only takes 4 hours when used. Thus our current implementation favors cases where the number of test samples is small. After training, the inference time of our method and unsup3d are the same (0.04s per image) as we adopt the same network architecture for shape prediction.
>
> It is also possible to modify our method to discard the need of instance-specific training by either biasing towards joint training or using the recovered shapes as ground-truths to train a global image-to-shape mapping network. But this is not the focus of this work, as the main value of this work is to reveal the geometric property of 2D GANs and its potential for 3D shape reconstruction.

---

### Decision · Program_Chairs · 2021-01-07
**Final Decision**

**Decision:**

Accept (Oral)

**Comment:**

The paper proposes to use pre-trained 2D (i.e., image) GANs as a mechanism for recovering 3D shape from a single 2D image. The work demonstrates impressive results on not only human and cat faces, but also cars and buildings. The method is demonstrated with qualitative results and quantitative results on multiple datasets and tasks.

The reviewers were persuaded by the novelty and "neatness" of the idea (and the AC is in agreement) as well as the results. At submission time, there were some concerns with experimental details. For instance, there was a question of how carefully the settings have to be tuned (always a concern with unsupervised methods) as well as an overarching concern about the initialization and whether the method will work on less clean data. The reviewers (and the AC) seem to think that these have been sorted out in discussion.

All three reviewers were in favor of acceptance and the area chair is inclined to agree with the reviewers. In particular, the AC finds the work interesting and compelling. While there is an updated version already uploaded during the discussion, the AC encourages the reviewers to double check all the questions from the reviewers and include the answers from the discussion into the camera ready (even these results are in the appendix).